# Risk Prediction Models for Peri-Operative Mortality in Patients Undergoing Major Vascular Surgery with Particular Focus on Ruptured Abdominal Aortic Aneurysms: A Scoping Review

**DOI:** 10.3390/jcm12175505

**Published:** 2023-08-24

**Authors:** Alessandro Grandi, Luca Bertoglio, Sandro Lepidi, Tilo Kölbel, Kevin Mani, Jacob Budtz-Lilly, Randall DeMartino, Salvatore Scali, Lydia Hanna, Nicola Troisi, Cristiano Calvagna, Mario D’Oria

**Affiliations:** 1Department of Vascular Medicine, University Heart and Vascular Center, 20251 Hamburg, Germany; 2Division of Vascular Surgery, Department of Clinical and Experimental Sciences, ASST Spedali Civili of Brescia, 25123 Brescia, Italy; 3Division of Vascular and Endovascular Surgery, Cardiovascular Department, University Hospital of Trieste ASUGI, 34129 Trieste, Italy; 4Section of Vascular Surgery, Department of Surgical Sciences, University of Uppsala, 751 05 Uppsala, Sweden; 5Division of Vascular Surgery, Department of Cardiovascular Surgery, Aarhus University Hospital, 8200 Aarhus, Denmark; 6Division of Vascular and Endovascular Surgery, Mayo Clinic, Rochester, MN 55905, USA; 7Division of Vascular Surgery and Endovascular Therapy, University of Florida, Gainesville, FL 32610, USA; 8Department of Surgery and Cancer, Imperial College London, London SW7 5NH, UK; 9Vascular Surgery Unit, Department of Translational Research and New Technologies in Medicine and Surgery, University of Pisa, 56126 Pisa, Italy

**Keywords:** vascular surgery, peri-operative mortality, risk models, scoping review, ruptured AAA

## Abstract

Purpose. The present scoping review aims to describe and analyze available clinical data on the most commonly reported risk prediction indices in vascular surgery for perioperative mortality, with a particular focus on ruptured abdominal aortic aneurysm (rAAA). Materials and Methods. A scoping review following the PRISMA Protocols Extension for Scoping Reviews was performed. Available full-text studies published in English in PubMed, Cochrane and EMBASE databases (last queried, 30 March 2023) were systematically reviewed and analyzed. The Population, Intervention, Comparison, Outcome (PICO) framework used to construct the search strings was the following: in patients with aortic pathologies, in particular rAAA (population), undergoing open or endovascular surgery (intervention), what different risk prediction models exist (comparison), and how well do they predict post-operative mortality (outcomes)? Results. The literature search and screening of all relevant abstracts revealed a total of 56 studies in the final qualitative synthesis. The main findings of the scoping review, grouped by the risk score that was investigated in the original studies, were synthetized without performing any formal meta-analysis. A total of nine risk scores for major vascular surgery or elective AAA, and 10 scores focusing on rAAA, were identified. Whilst there were several validation studies suggesting that most risk scores performed adequately in the setting of rAAA, none reached 100% accuracy. The Glasgow aneurysm score, ERAS and Vancouver score risk scores were more frequently included in validation studies and were more often used in secondary studies. Unfortunately, the published literature presents a heterogenicity of results in the validation studies comparing the different risk scores. To date, no risk score has been endorsed by any of the vascular surgery societies. Conclusions. The use of risk scores in any complex surgery can have multiple advantages, especially when dealing with emergent cases, since they can inform perioperative decision making, patient and family discussions, and post hoc case-mix adjustments. Although a variety of different rAAA risk prediction tools have been published to date, none are superior to others based on this review. The heterogeneity of the variables used in the different scores impairs comparative analysis which represents a major limitation to understanding which risk score may be the “best” in contemporary practice. Future developments in artificial intelligence may further assist surgical decision making in predicting post-operative adverse events.

## 1. Introduction

There is general agreement that clinical decision aids improve patient knowledge, foster more realistic expectations, mitigate decisional conflicts and reduce the proportion of patients unable to reach a verdict when considering medical decisions [1]. To support good clinical judgement, a reliable decision tool needs to be based both clinically relevant and feasible to use in a dynamic clinical setting. This is especially true in patients undergoing vascular surgery who are at an elevated risk for both perioperative complications and long-term mortality due to their age and multiple comorbidities [2].

Different risk calculators and clinical tools have been studied and developed to evaluate the impact of chronic comorbidities on outcomes following surgery. These include the Revised Cardiac Risk Index (RCRI), [3] the Charlson Comorbidity Index (CCI), [4] age-adjusted CCI, [5] and the American Society of Anesthesiologists (ASA) Physical Status Classification System [6]. Furthermore, current American Heart Association guidelines highlight the elevated surgical risk independent of patient-related risk factors carried in different vascular surgery procedures [7]. The National Surgery Quality Improvement Program risk calculator [8] and the Vascular Quality Initiative Cardiac Risk Predictor [9] are two of the contemporary risk calculators that have tried to increase the accuracy of risk scoring by accounting for the type of procedure. However, these calculators, while providing only a risk assessment for the perioperative period, require the input of data from supplemental testing and are therefore not suitable for longer-term prediction, including mid- and long-term mortality. Because of this, there is a limit to the ease of using existing calculators and their utility in assessment of procedure appropriateness relating to both perioperative risk and late mortality.

Furthermore, precise risk scores can aid physicians in emergent situations such as the decisions surrounding treatment of a ruptured abdominal aortic aneurysm (rAAA). Finally, precise risk scores can be used to evaluate clinical outcomes while also enhancing the ability to perform center-level comparisons when attempting to benchmark results for quality improvement efforts. To date, a variety of different risk scores have been reported across the spectrum of vascular procedures; however, it remains unclear which decision aid tool is optimal, especially when confronted with emergency presentations such as an rAAA.

Therefore, the present scoping review aims to describe and analyze available clinical data on the most frequently reported risk prediction indices in vascular surgery for perioperative mortality, with a particular focus on rAAA, offering a narrative review to capture salient themes and gaps in the literature, thus providing a clearer pathway for future research.

## 2. Materials and Methods

### 2.1. Study Design

A scoping review following the PRISMA Protocols Extension for Scoping Reviews was performed [10] (Appendix A). Available full-text studies published in English in PubMed, Cochrane and EMBASE databases were systematically reviewed and analyzed (last queried, 30 March 2023). Reference lists from all included manuscripts were manually screened and included if necessary. The Population, Intervention, Comparison, Outcome (PICO) framework used to construct the search strings was the following (Appendix A): in patients with aortic pathologies, in particular rAAA (population), undergoing open or endovascular surgery (intervention), what different risk prediction models exist (comparison), and how well do they predict post-operative mortality (outcomes)? Duplicate copies of articles were identified and removed. Case reports, letters, editorials, commentaries and manuscripts written in a language other than English were excluded.

### 2.2. Data Extraction and Evidence Synthesis

Data were reported as descriptive narrative or tables, without any statistical analysis or quality assessment of the included papers, in accordance with the PRISMA guidelines for scoping reviews. Data extraction was performed using Microsoft Excel software. Two authors (A.G., M.D.) independently assessed the studies for inclusion in the review; in case of disagreement, a third author (L.B.) was involved to achieve consensus. The following data were extracted: list of authors, publication year, number of patients in the study, risk score details used, and perioperative complications rate.

## 3. Results

### 3.1. General Overview

After the literature search and screening of all relevant abstracts, a total of 56 studies were included in the final qualitative synthesis (Figure 1).

The main findings of the scoping review, grouped by the risk score that was investigated in the original studies, are reported in detail in the following paragraphs.

### 3.2. Perioperative Mortality Risk Scores in Major Vascular Surgery/Elective AAA (Table 1 and Table 2)

#### 3.2.1. Comorbidity–Polypharmacy Score (CPPS)

The comorbidity–polypharmacy score (CPPS) was originally developed in an emergency department setting in order to rapidly quantify the cumulative comorbid condition severity in trauma patients >45 years of age [11]. It is calculated as the sum of the number of preinjury comorbid conditions and home medications and is classified into four groups of severity: 0–7 (mild), 8–14 (moderate), 15–21 (severe) and >21 (morbid) [12,13]. CPPS has been used to predict risk of short-term mortality and in-hospital complications [14,15,16], as well as other short- and long-term outcomes [13,14,15,16,17], including 1-year survival [18]. CPPS has been shown to be comparable with CCI in predicting outcomes in trauma patients >45 years of age [14,18]. Khanh et al. [19] applied the CPPS to a total of 466 patients undergoing vascular surgery, 61 of which underwent endovascular aortic aneurysm repair. A higher CPPS was associated with a longer hospital stay (*p* < 0.001). CPPS was independently associated with 1- and 5-year mortality in a multivariable Cox model [hazard ratio (HR): 2.2, 95% confidence interval (CI): 1.3–3.3]. The receiver operator characteristic (ROC) yielded C-statistics of 0.81 and 0.72 for 1-year and 5-year all-cause mortality, respectively (*p* < 0.001).

#### 3.2.2. Long-Term Survival Score (LTSS)

The long-term survival score (LTSS) proposed by Landsberg et al. [20] is composed of the RCRI criteria (congestive heart failure, ischemic heart disease, insulin-treated diabetes mellitus, chronic renal failure and cerebrovascular disease), with the addition of age >65 years, ST-segment depression on preoperative 12-lead electrocardiography (ECG), and both insulin-treated and insulin-independent diabetes mellitus. This score is used to predict survival in the long run after major vascular surgery. In their study, patients were stratified according to the number of met criteria. Low-risk patients had good long-term survival and did not require preoperative coronary revascularization. Conversely, high-risk patients had worse long-term survival, which did not show improvement with preoperative cardiac testing or coronary revascularization. Notably, those who scored in the intermediate-risk group had improved long-term survival with preoperative coronary revascularization. LTSS was further validated by Subramaniam et al. [21] in a cohort of 921 consecutive patients undergoing major vascular surgery from two institutions. The predicted mortality of the LTSS was more precise than the RCRI (ROC curves were at six months (0.66 ± 0.03 vs. 0.57 ± 0.04, *p* = 0.02) and three years (0.70 ± 0.02 vs. 0.61 ± 0.02, *p* < 0.0001)) in both institutions.

#### 3.2.3. Simple Vascular Quality Initiative-Frailty Score (VQI-FS)

Kraiss et al. [22] proposed an abbreviated frailty score (the Vascular Quality Initiative-Frailty Score (VQI-FS)) that was developed using 11 variables in order to recognize frailty domains in the Comprehensive Geriatric Assessment. Non-emergent cases registered in the SVS Vascular Quality Initiative from 2010 to 2017 (n = 265,632) in seven registries (carotid endarterectomy, n = 77,111; carotid artery stenting, n = 13,215; endovascular abdominal aortic aneurysm repair, n = 29,607; open AAA repair (OAR), n = 7442; infrainguinal bypass, n = 33,128; suprainguinal bypass, n = 10,661; peripheral vascular intervention, n = 94,468) were analyzed using logistic regression models to determine the predictive power of the VQI-FS for perioperative and post-discharge mortality. The VQI-FS, using equal weighting of these 11 VQI variables, effectively predicted 9-month mortality with an ROC value of 0.724. The model calibration was excellent, with predicted/observed regression line slope (0.991) and intercept (5.449 × 10^−4^).

#### 3.2.4. Physiological and Operative Severity Score for the enumeration of Mortality and Morbidity (POSSUM)

In 1991, Copeland et al. [23] created the Physiological and Operative Severity Score for the enUmeration of Mortality and Morbidity (POSSUM). Over the years, new versions of the score have been developed, such as Portsmouth-POSSUM (P-POSSUM) by Prytherch et al. [24,25] and Vascular-POSSUM (V-POSSUM) by the VSGBI [26]. The V-POSSUM physiology score is derived from the V-POSSUM and considers only preoperative data [27] and V-POSSUM Cambridge, designed by Tang et al. [28]. POSSUM scoring systems are widely used and validated tools for 30-day mortality [29] and morbidity prediction. 

Byrne et al. [30] evaluated the V-POSSUM score in 106 patients. Predicted and observed morbidity (41% and 35.8%, respectively) were not significantly different (*p* = 0.066). V-POSSUM did, however, over-predict mortality (9.7 vs. 5.7%; *p* = 0.021). While the discrimination for predicting morbidity was poor, V-POSSUM scores were significantly associated with mortality endpoints (area under the ROC curve = 0.97250). V-POSSUM morbidity scores closely correlate with observed outcomes.

Midwinter et al. [31] tested the POSSUM and P-POSSUM scores in 221 vascular surgery patients. The study found that the POSSUM score overestimated deaths, while the P-POSSUM score was not significantly different from the observed death rate. Also, the risk of morbidity predicted by the POSSUM was not significantly different from the observed complication rate. 

Those five scores were recently compared in 208 elderly patients (≥65 years) undergoing major vascular surgery by Teixeira et al. [32]. The Hosmer–Lemeshow test (H-L T) and Standardized Mortality/Morbidity Ratio (SMR) were used to assess the POSSUM system’s goodness-of-fit for predicting mortality and morbidity and the discriminative ability by ROC curves. The patients’ average age was 70.8 years, with 81% males. At 30 days, 6 patients died (2.97%), and 59 had at least one complication (29.2%). The predicted overall mortality for POSSUM, P-POSSUM, V-POSSUM, V-POSSUM physiology and V-POSSUM Cambridge equation was 29.1, 4.43, 15.3, 21.9 and 13.5 deaths, respectively. One hundred and five complications were predicted by the equation for POSUSM morbidity. The *p*-values for the H-LT were 0.001, 0.164, 0.208, 0.011, 0.331 and 0.001, respectively, while SMRs were 0.21 (0.04–0.37), 1.35 (0.27–2.44), 0.39 (0.08–0.71), 0.27 (0.06–0.49), 0.44 (0.09–0.80) and 0.56 (0.42–0.71), respectively, and ROC values were 0.72 (0.49–0.95), 0.72 (0.49–0.95), 0.73 (0.51–0.94), 0.69 (0.50–0.89), 0.72 (0.52–0.92) and 0.71 (0.63–0.79), respectively. The authors concluded that the best performance predicting 30-day mortality was performed by the P-POSSUM score. All the other scores overestimated mortality at 30 days. Furthermore, the prediction of morbidity was inadequate. For elderly patients undergoing major elective vascular surgery, to date, the POSSUM scoring models may not be suitable enough for risk prediction as further calibration and discrimination is required. 

#### 3.2.5. The New Zealand Vascular Surgical Risk Tool (NZRISK-VASC)

Kim et al. [33] recently proposed and validated the New Zealand Vascular Surgical Risk Tool (NZRISK-VASC) in 21.597 vascular surgery patients. The authors formed the final risk prediction models including gender, urgency, ASA score, cancer status, renal failure, diabetes, anatomical site, structure operated and endovascular procedure. The ROC value for 30-day, one-, and two-year mortality using the L-min model was 0.869, 0.833, and 0.824, respectively, demonstrating very good discrimination. Calibration with the validation dataset was also excellent, with slopes of 0.971, 1.129, and 1.011, respectively, and McFadden’s pseudo-R2 statistics of 0.250, 0.227, and 0.227, respectively.

#### 3.2.6. Preoperative Score to Predict Postoperative Mortality (POSPOM)

In 2016, Le Manach et al. [34] introduced the Preoperative Score to Predict Postoperative Mortality (POSPOM). The score is derived from surgical procedures requiring anesthesia, covering emergency as well as elective operations, as well as patients’ age and significant comorbidities. The latter information is used to calculate an individual score value, indicating the patient’s risk for postsurgical in-hospital death. The POSPOM was derived from data of French hospitals where at least 500 surgical procedures were performed in adults within the year 2010. By involving more than 5.5 million patient datasets in either a derivation or validation cohort, Le Manach et al. generated and validated a convincing prognostic model. The POSPOM was further validated in Germany by Layer et al. [35] using data from 199,780 surgical cases. POSPOM showed a good performance with a c-statistic of 0.771 and a Brier score of 0.021. Furthermore, Reis et al. [36] compared POSSUM and POSPOM to different ICU risk scores to predict mortality in 833 patients admitted to ICU after open vascular surgery. The observed mortality was within the predicted range (1–5% after intermediate-risk and >5% after high-risk surgery). 

**Table 1 jcm-12-05505-t001:** Perioperative mortality risk scores in major vascular surgery.

Risk Score	Author, Year	Algorithm/Variables	Prediction
Comorbidity–Polypharmacy Score (CCPS)	Evans, 2012 [13]	Sum of the number of preinjury comorbid conditions and home medications	0–7 (mild), 8–14 (moderate), 15–21 (severe) and >21 (morbid)
Long-term survival score (LTSS)	Landsberg, 2006 [20]	RCRI criteria (congestive heart failure, ischemic heart disease, insulin-treated diabetes mellitus, chronic renal failure and cerebrovascular disease), +age > 65 years, ST-segment depression on preoperative 12-lead EKG, and both insulin-treated and insulin-independent DM	0–1 (low risk), 2–3 (intermediate risk) and ≥4 (high risk)
Simple Vascular Quality Initiative-Frailty Score (VQI-FS)	Kraiss, 2022 [22]	Congestive heart failure, renal impairment, chronic obstructive pulmonary disease, not living at home, not ambulatory, anemia and underweight status	%
Physiological and Operative Severity Score for the enumeration of Mortality and Morbidity (POSSUM)	Copeland, 1991 [23]	Age, cardiac, respiratory, BP, HR, GCS, HB, WBC, Urea, Sodium, Potassium, EKG, operative severity, number of procedures, EBL, peritoneal soiling, malignancy and urgency	%
The New Zealand Vascular Surgical Risk Tool (NZRISK-VASC)	Kim, 2021 [33]	ASA score, gender, urgency, malignancy, presence of renal failure, diabetes, anatomical site, structure operated and endovascular procedure	%
Preoperative Score to Predict Postoperative Mortality (POSPOM)	Le Manach, 2016 [34]	Age, ischemic heart disease, cardiac arrhythmia or heart blocks, chronic heart failure or cardiomyopathy, peripheral vascular disease, dementia, cerebrovascular disease, hemiplegia, chronic obstructive pulmonary disease, chronic respiratory failure, chronic alcohol abuse, cancer, diabetes, transplanted organ(s), chronic dialysis, chronic renal failure, and type of surgery	0–50. ≤20: ≤0.04% 25: 1.73% 30: 5.65% 40: 11.77%
British Aneurysm Repair score (BAR)	Grant, 2013 [37]	Open repair, increasing age, female sex, serum creatinine level over 120 µmol/L, cardiac disease, abnormal electrocardiogram, previous aortic surgery or stent, abnormal white cell count, abnormal serum sodium level, AAA diameter and ASA fitness grade	Low risk: 0.8% Medium risk: 2.3% High risk: 7.1%

RCRI, Revised Cardiac Risk Index; EKG, electrocardiogram; BP, blood pressure; HR, heart rate; GCS, Glasgow Coma Scale; HB, hemoglobin; WBC, white blood count; EBL, estimated blood loss; ASA, American Society of Anesthesiologists.

**Table 2 jcm-12-05505-t002:** Clinical validation literature on perioperative mortality risk scores in major vascular surgery.

Author, Year	Score	n	Score Association	Mortality Association	Mortality ROC Value
Khanh, 2020 [19]	CPPS	466 (61 EVAR)	longer LOS (*p* < 0.001)	HR 2.2, CI 1.3–3.3	1-year value: 0.81 5-year value: 0.72 (*p* < 0.001)
Subramaniam, 2011 [21]	LTSS vs. RCRI	921	LTSS provides better discrimination between each adjacent two-risk score than RCRI	LTSS provides a better prediction than RCRI	6-month value: 0.66 ± 0.03 vs. 0.57 ± 0.04, *p* = 0.02 3-year value: 0.70 ± 0.02 vs. 0.61 ± 0.02, *p* < 0.0001
Byrne, 2009 [30]	V-POSSUM	106	Predicted and observed morbidity (41 and 35.8%, respectively) were not significantly different (*p* = 0.066). V-POSSUM morbidity scores closely correlate with observed outcomes.	Significant association	0.97250
Midwinter, 1999 [31]	POSSUM and P-POSSUM	221	The risk of morbidity predicted by the POSSUM was not significantly different from the observed complication rate.	POSSUM score overestimated deaths, while the P-POSSUM score was not significantly different from the observed death rate.	-
Teixeira, 2018 [32]	POSSUM, P-POSSUM, V-POSSUM, V-POSSUM physiology, and V-POSSUM Cambridge equation	208 (≥65 years)	Prediction of morbidity was inadequate.	P-POSSUM had the best performance when predicting 30-day mortality. All the others overestimated 30-day mortality.	0.21 [0.04–0.37], 1.35 [0.27–2.44], 0.39 [0.08–0.71], 0.27 [0.06–0.49], 0.44 [0.09–0.80] and 0.56 [0.42–0.71]
Layer, 2021 [35]	POSPOM	199,780	-	Good performance	0.771
Reis, 2019 [36]	POSSUM and POSPOM	833 ICU patients	-	Observed mortality was within the predicted range (1–5% after intermediate-risk and >5% after high-risk surgery). POSSUM and POSPOM had slightly better predictive capacity than the ICU risk scores.	-

ICU, intensive care unit.

#### 3.2.7. British Aneurysm Repair Score (BAR)

Proposed by Grant et al. [37] in 2013, this score focuses on mortality after elective OAR and endovascular AAA repair (EVAR). The authors analyzed 11,423 AAA with 312 in-hospital deaths [2.7% (2.4–3.0)]: 230 after 4940 OAR [4.7% (4.1–5.3)] and 82 after 6483 EVAR [1.3% (1.0–1.6)]. The following variables were included in the final model as they were associated with in-hospital death: OAR, increasing age, female sex, serum creatinine level over 120 µmol/L, cardiac disease, abnormal electrocardiogram, previous aortic surgery or stent, abnormal white cell count, abnormal serum sodium level, AAA diameter and ASA fitness grade. The ROC value was 0.781 (CI. 0.756–0.806) with a bias-corrected value of 0.774. Model calibration was good (*p* = 0.963) based on the H-L T goodness-of-fit test, (bias-corrected) calibration curves, risk group assessment and recalibration regression. The authors concluded that the presented multivariable model for elective AAA repair can be used to risk-adjust outcome analyses and provide patient-specific estimates of in-hospital mortality risk for OAR or EVAR.

### 3.3. Perioperative Mortality Risk Scores in Ruptured Abdominal Aortic Aneurysm (rAAA) (Table 3 and Table 4)

#### 3.3.1. Hardmann Index (HI)

Proposed by Hardman et al. [38], the Hardman index (HI) scoring system uses five variables collected preoperatively on admission, worth one point each. A score ≥ 3 indicates high mortality risk. These variables are as follows: age > 76 years, serum creatinine > 190 μmol/L, hemoglobin < 9 g/dL, episode of loss of consciousness after presentation, and evidence of cardiac ischemia (>1 mm ST segment depression or associated T-wave change) on ECG. The authors validated their scoring system by analyzing 154 patients with rAAA. The hospital mortality rate was 39%. Patients with a single risk factor (n = 52) had a mortality rate of 37%; those with two factors (n = 32) had a mortality rate of 72%; those with three or more factors (n = 8) had a mortality rate of 100%; and no patient had all five risk factors. The patients with no risk factors (n = 62) presented a mortality rate of 16%.

Conroy et al. [39] retrospectively studied 95 emergent EVAR patients for rAAA. The mortality rates at 24 h and 30 days were 16% and 36%, respectively. It was found that an increased HI score was directly correlated to increased mortality. The authors concluded that the HI can predict an increased risk of 30-day mortality from endovascular repairs of rAAA. However, mortality from endovascular repair is much lower than would be predicted in OAR, and it therefore cannot be used clinically as a tool for exclusion from intervention.

#### 3.3.2. Glasgow Aneurysm Score (GAS)

The Glasgow Aneurysm Score (GAS), designed in 1994 by Samy et al. [40] for predicting outcomes after repair of intact or rAAA, calculates risk of death on the basis of patient age, preoperative shock, and myocardial, cerebrovascular and renal disease by analyzing 500 patients. Through regression coefficients, the authors postulated the following equation: risk score = (age in years) + (17 for shock) + (7 for myocardial disease) + (10 for cerebrovascular disease) + (14 for renal disease). A score >95 indicates high mortality risk (>80%). Thanks to its simplicity, GAS has been used for case-mix assessment in international quality registries, as the variables required for GAS are often readily available in quality registries [41].

The GAS was further validated by different authors. Özen et al. [42] analyzed 121 patients diagnosed with rAAA who underwent OAR. The reported perioperative death rate was 39.7% (n = 48). The GAS was 84.15 ± 15.94 in the group of patients who died and 75.14 ± 14.67 in the group of patients who survived (*p* = 0.002). The authors concluded that the GAS may have a predictive outcome value in patients with rAAA undergoing OAR, and when integrated with clinical experience, it could help the individual patient decision making process. Korhonen et al. [43] assessed 836 patients who underwent surgery for rAAA. Of those, 395 (47.2%) died in the perioperative period, 164 (19.6%) suffered cardiac complications and 164 (19.6%) needed ICU stay >five days. Through univariate analysis, CAD (*p* = 0.005), preoperative shock (*p* < 0.001), age (*p* < 0.001) and the GAS (*p* < 0.001) were found to be mortality predictors: preoperative shock [odds ratio [OR] 2.13 (CI 1.45–3.11); *p* < 0.001] and the GAS [for an increase of ten units: OR 1.81 (CI 1.54–2.12); *p* < 0.001] were found to be independently associated with mortality. The ROC cut-off value for mortality prediction with GAS was 84 [0.75 (CI 0.72–0.78), SD 0.17; *p* < 0.001]. The operative mortality rate was 28.2% (114 of 404) in patients with a GAS ≤ 84, compared with 65% (281 of 432) in those with a GAS > 84 (*p* < 0.001). The authors concluded that the GAS predicted postoperative death after OAR of rAAA in their series. 

More recently, the GAS has been updated to include the type of operation performed by Visser et al. [44]. The updated formula is as follows: age (years) + 7 for cardiac comorbidity (defined as previous history of myocardial infarction, cardiac surgery, angina pectoris or arrhythmia) + 10 for cerebrovascular comorbidity (defined as previous history of stroke or transient ischemic attack) + 17 for shock (defined as an in hospital systolic blood pressure < 80 mmHg) + 14 for renal insufficiency (defined as a pre-operative serum creatinine > 160 mmol/L) + 7 for OAR.

#### 3.3.3. Vancouver Score

Proposed by Chen et al. [45], the Vancouver score is probably the least well known and used. The original retrospective study examined 147 patients; by using multivariate logistic regression analysis, the authors were able to identify age, reduced consciousness, and preoperative cardiac arrest as significant predictors of death. These variables could be entered into a predictive model, and the probability of death was estimated using the equation [e^x^/(1 + e^x^)], where *e* is the base of the natural logarithm, and *x* = −3.44 + age (years) × 0.062 + loss of consciousness (yes = 1; no = −1) × 1.14 + cardiac arrest (yes = 1; no = −1) × 0.6. The outcome of the formula is the mortality risk. The authors also prospectively validated their formula in a subsequent cohort of 134 patients [46]. They argue that their system was accurate at predicting patients at extreme risk (patients with a predicted mortality > 90%); however, the model fared less satisfactorily for patients with a predicted mortality < 80%). 

#### 3.3.4. Edinburgh Ruptured Aneurysm Score (ERAS)

Proposed by Tambyraja et al. [47], the Edinburgh Ruptured Aneurysm Score (ERAS) was validated on 105 patients treated for rAAA. At 30 days, there were 39 (37%) deaths. Hemoglobin < 9 g/dl (*p* = 0.038), blood pressure < 90 mmHg (*p* = 0.036), and Glasgow Coma Scale (GCS) < 15 (*p* = 0.016) were found to be mortality risk factors at univariate analysis. The mortality rate was as follows: in patients with no or one risk factor, 29% (20/70); in patients with two factors, 50% (15/30); in patients with all three factors, 80% (4/5). A correlation between cumulative risk factors and mortality was found (*p* = 0.003).

Therefore, a point was awarded for GCS score < 15, systolic blood pressure < 90 mmHg, and preoperative hemoglobin level < 5.6 mmol/L. A score ≤ 1 indicates 30% mortality, a score = 2 indicates 50% mortality and a score = 3 indicates 80% mortality. 

The ERAS has been further validated and compared to other risk score in a prospective study [48]. During the study period, 111 patients were admitted with rAAA. Of these, 84 (76%) attempted OAR and were included in the study; 37 (44%) died after operation. While the V-POSSUM equation effectively predicted mortality (*p* = 0.086), there was a lack of fit for the POSSUM derivative (*p* = 0.009). The authors concluded that the retrospective validation shows that the HI, GAS, and V-POSSUM and POSSUM scores do not perform well as predictors for death after rAAA. The ERAS accurately stratifies perioperative risk but requires further validation.

#### 3.3.5. Vascular Study Group of New England (VSGNE) rAAA

Proposed by Robinson et al. [49], this scoring system was studied in the United States through the Vascular Study Group of New England (VSGNE) registry. During the study period, 242 patients underwent OAR of rAAAs at 10 centers. Independent predictors of mortality included age >76 years (OR 5.3; CI 2.8–10.1), preoperative cardiac arrest (OR 4.3; CI 1.6–12), loss of consciousness (OR 2.6; CI 1.2–6) and suprarenal aortic clamp (OR 2.4; CI 1.3–4.6). Patient stratification according to the VSGNE RAAA risk score (range 0–6) accurately predicted mortality and identified those at low and high risk for death (8%, 25%, 37%, 60%, 80% and 87% for scores of 0, 1, 2, 3, 4 and 5, respectively). Discrimination (C = 0.79) and calibration (χ^2^ = 1.96; *p* = 0.85) were excellent in the derivation and bootstrap samples and superior to that of existing scoring systems. 

#### 3.3.6. Rapid Ruptured Abdominal Aortic Aneurysm Score (RrAAAS)

Presented by Healey et al. [50] as an update to the VSGNE rAAA, the Rapid Ruptured Abdominal Aortic Aneurysm Score (RrAAAS) analyzed 649 patients who underwent repair of rAAA; of these, 247 (38.1%) underwent EVAR, and 402 (61.9%) underwent OAR. On multivariate modeling, the primary determinants of mortality at 30 days were advanced age (>76 vs. ≤76 years, OR 2.91 and CI 2.0–4.24), elevated creatinine (>1.5 mg/dL vs. ≤1.5 mg/dL, OR 1.57 and CI 1.05–2.34) and the lowest systolic blood pressure (SBP) (BP <70 mmHg vs. ≥70 mmHg, OR 2.65 and CI 1.79–3.92). The logistic regression model had an ROC value of 0.69. The corresponding linear model used to provide a point estimate of 30-day mortality (%) was as follows: % mortality = 14 + 22 × (age > 76) + 9 × (creatinine > 1.5) + 20 × (bp < 70). Using this model, patients can be stratified into different groups, each with a specific estimated risk of mortality at 30 days ranging from 14% to 65%.

This model was later externally validated by Neilson et al. [51], who analyzed the VQI registry containing 2704 eligible patients, of which 715 had been contributed by VSGNE. The discrimination of RrAAAS was similar to GAS or ERAS (ROC0.66). Neither GAS nor ERAS provides a direct prediction of mortality; observed mortality in the VQI minus VSGNE cohort tended to be somewhat lower than predictions of the original RrAAAS. A recalibrated equation predicting the percent mortality was as follows: Mortality (%) = 16 + 12 × (age > 76) + 8 × (creatinine > 1.5) + 20 × (systolic blood pressure < 70). The authors concluded that the previously described RrAAAS has a similar discrimination as the GAS and ERAS, is easier to obtain in an emergency setting and has been recalibrated to reflect the experience of a large national sample. 

#### 3.3.7. Dutch Aneurysm Score (DAS)

Proposed by von Meijenfeldt et al. [52], the Dutch Aneurysm Score (DAS) was developed using a multivariate logistic regression model on a prospective cohort of 508 patients from 10 different hospitals as well as externally validated using two retrospective cohorts of 737 rAAA patients from two different hospitals. Age, lowest in-hospital systolic blood pressure, cardiopulmonary resuscitation and hemoglobin level were identified to be associated with mortality. The ROC was compared with the GAS (0.77, CI 0.72–0.82 vs. 0.72, CI 0.67–0.77). The authors were able to use this score to show an 83% mortality in patients with a predicted death rate ≥80%.

#### 3.3.8. Clinical Assessment of Instability—Weingarten Score

Weingarten et al. [53] examined 125 patients (40 stable) and compared the association of presenting clinical acuity, defined an unstable patient as an individual with an rAAA and profound hypotension, preoperative cardiac arrest, loss of consciousness, and/or the need for preoperative tracheal intubation, and GAS. Therefore, this score may be qualified as a refined version of the GAS. The perioperative mortality rate for unstable and stable cases were 41% and 12%, respectively. (*p* < 0.001) The sensitivity and specificity of the unstable status for perioperative mortality were 88% and 41%, respectively. The authors described a direct correlation between higher GAS and perioperative mortality (*p* = 0.001). With ROC analysis (0.72, CI 0.62–0.82) a GAS cut-off of 96 was found to have 63% and 72% sensitivity and specificity, respectively. The perioperative mortality was 51% (25/49) for patients above this cut-off and 20% (15/76) for patients below it. Stable and unstable patients had an estimated one-year survival of 75% (CI 62–91%) and 48% (CI 38–60%), respectively. For patients above and below the GAS cut-off, the estimated 1-year survival was 23% (CI 13–40%) and 77% (CI 67–87%), respectively. In conclusion, through clinical presentation and GAS the authors could preemptively identify rAAA patients with a high predicted mortality. The identified GAS cut-off was able to identify patients with poor long-term survival, although 42% of these patients survived one year. This showed that those indicators were not helpful in understanding the futility of surgery.

The same score was used by Jàcome et al. [54] to compare its prognostic validity with the GAS and the Vancouver Scoring System in 120 patients. The authors showed no superiority in perioperative mortality prediction for rAAA patients. Furthermore, they confirmed the inability of the scores to predict the futility of intervention. 

#### 3.3.9. Artificial Neuronal Network (ANN)

Proposed by Wise et al. [55], an artificial neuronal network (ANN) to analyze 125 patients undergoing EVAR or OAR for rAAA with a reported mortality rate of 42% (n = 53) Five independent preoperative factors were associated with perioperative mortality age ≥ 70, renal disease, loss of consciousness, cardiac arrest and shock, although renal disease was excluded from the models. The presence of any risk factor increased mortality from 11% to 16% to 44% to 76% to 89%. Algorithms derived from multiple logistic regression, ANN, and GAS models generated ROC values of 0.85 ± 0.04, 0.88 ± 0.04 (training set), and 0.77 ± 0.06, respectively, and Pearson r^2^ values of 0.36, 0.52 and 0.17, respectively. The most discriminant model was found to be the ANN. The authors concluded that this predictive model could help physicians identify rAAA patients with a high perioperative mortality risk.

#### 3.3.10. Harborview Medical Center Preoperative Risk Score (HRS)

Garland et al. [56] analyzed 303 patients, 16 of which died before undergoing the operation. Independent preoperative variables associated mortality were age > 76 (OR 2.11; CI 1.47–4.97; *p* = 0.011), creatinine concentration > 2.0 mg/dL (OR 3.66; CI 1.85–7.24; *p* < 0.001), pH < 7.2 (OR 2.58; CI 1.27–5.24; *p* = 0.009) and systolic blood pressure ever < 70 mmHg (OR 2.70; CI 1.46–4.97; *p* = 0.002). Patients were stratified according to the number of risk factors (range 0–4). At 30 days, the mortality rates were 22% for patients with one point, 69% for two points, and 80% for three points. No patient with four points survived. Using EVAR in rAAA patients showed a mortality reduction in all score categories. The authors concluded that the presented rAAA mortality risk score named Harborview Medical Center preoperative risk score (HRS) has the advantage of using easily assessed variables, while facilitating accurate predictions. Furthermore, it helps physicians with their clinical decision making and in discussions with patients and their families.

The HRS has been recently validated with a prospective cohort [57] of 118 patients [45 (38.1%) OAR, 61 (51.7%) EVAR, and 12 (10.2%) no intervention]. In the operated patients, a significant linear trend was shown between the HRS and perioperative death for all patients (*p* < 0.0001), for OAR (*p* = 0.0003), and for EVAR (*p* < 0.0001). For all repairs, a score of 0 was associated with a 14.6% mortality rate, a score of one a 35.7% mortality rate, a score of two a 68.4% mortality rate, and a score of three and four a 100% mortality rate.

**Table 3 jcm-12-05505-t003:** Perioperative mortality risk scores in ruptured abdominal aortic aneurysms.

Risk Score	Author, Year	Algorithm/Variables	Mortality Prediction
Hardmann Index	Hardman, 1996 [38]	Age > 76 years, serum creatinine >190 μmol/L, HB < 9 g/dL, episode of loss of consciousness (defined as any syncopal episodes), and evidence of cardiac ischemia (>1 mm ST segment depression or associated T-wave change) on EKG	0 factors: 16% 1 factor: 37% 2 factors; 72% ≥3 factors: 100% No patients had all 5.
Glasgow aneurysm score (GAS)	Samy, 1994 [40]	Risk score = (age in years) + (17 for shock) + (7 for myocardial disease) + (10 for cerebrovascular disease) + (14 for renal disease).	>95 = >80%
Updated Glasgow aneuyrys score	Visser, 2004 [44]	Age (years) + 7 for cardiac comorbidity (defined as previous history of myocardial infarction, cardiac surgery, angina pectoris or arrhythmia) + 10 for cerebrovascular comorbidity (defined as previous history of stroke or transient ischemic attack) + 17 for shock (defined as an in hospital systolic blood pressure <80 mmHg) + 14 for renal insufficiency (defined as a pre-operative serum creatinine >160 mmol/L) + 7 for OAR	%
Vancouver score	Chen, 1996 [45]	[e^x^/(1 + e^x^)], where *e* is the base of the natural logarithm and *x* = −3.44 + age (years) × 0.062 + loss of Consciousness (yes = 1; no = −1) × 1.14 + cardiac arrest (yes = 1; no = −1) × 0.6	%
Edimburgh Ruptured Aneurysm Score (ERAS)	Tambyraja, 2007 [47]	GCS < 15, systolic BP < 90 mmHg, and HB < 5.6 mmol/L	Score ≤ 1 = 30% Score = 2 = 50% Score = 3 = 80%
Vascular Study Group Of New England (VSGNE) rAAA	Robinson, 2009 [49]	Age > 76 years (OR 5.3; CI 2.8–10.1), preoperative cardiac arrest (OR 4.3; CI 1.6–12), loss of consciousness (OR 2.6; CI 1.2–6), and suprarenal aortic clamp (OR 2.4; CI 1.3–4.6).	0 = 8% 1 = 25% 2 = 37% 3 = 60% 4 = 80% 5 = 87%
Rapid Ruptured Abdominal Aortic Aneurysm Score (RrAAAS)	Healey, 2017 [50]	% mortality = 14 + 22 × (age >76) + 9 × (creatinine >1.5) + 20 × (bp <70)	14–65%
Dutch aneurysm score (DAS)	von Meijenfeldt, 2017 [52]	Age, lowest in-hospital systolic blood pressure, cardiopulmonary resuscitation, and hemoglobin level	≥80% = 83%
Weigarten score	Weigarten, 2015 [53]	Unstable status: hypotension, preoperative cardiac arrest, loss of consciousness, and/or the need for preoperative tracheal intubation	-
Artificial Neuronal Network (ANN)	Wise, 2015 [55]	Age ≥ 70, loss of consciousness, cardiac arrest, and shock	0 = 11% 1 = 16% 2 = 44% 3 = 76% 4 = 89%
Harborview Medical Center preoperative risk score (HRS)	Garland, 2017 [56]	Age >76 years (OR 2.11; CI 1.47–4.97; *p* = 0.011), creatinine concentration >2.0 mg/dL (OR 3.66; CI 1.85–7.24; *p* < 0.001), pH <7.2 (OR 2.58; CI 1.27–5.24; *p* = 0.009), and systolic blood pressure ever <70 mmHg (OR 2.70; CI 1.46–4.97; *p* = 0.002)	1 = 22% 2 = 69% 3 = 80% 4 = 100%

HB, hemoglobin; GCS, Glasgow coma scale; BP, blood pressure; HB, hemoglobin.

**Table 4 jcm-12-05505-t004:** Clinical validation literature on perioperative mortality risk scores in ruptured abdominal aortic aneurysms.

Author, Year	Score	n	Mortality Association
Conroy, 2020 [39]	Harmann index	95 EVAR	Increasing scores on the Hardman index showed an increasing mortality rate. Thirty-day mortality score 0–2 = 30.5%; score ≥ 3 = 69.2% (*p* = 0.01, RR 2.26, CI 0.98–5.17). This is lower than predicted in both patient groups based on the Hardman index score. Loss of consciousness was the only statistically significant independent predictor of 30-day mortality with a risk ratio of 3.16 (CI 2.00–4.97, *p* < 0.001).
Özen, 2015 [42]	GAS	121 OR	The most appropriate cut-off value for GAS was determined as 78.5 (AUC = 0.669, *p* = 0.002, sensitivity: 64.6%, specificity: 60.3%). GAS value above 78.5 is associated with almost threefold increase in mortality (*p* = 0.007, OR:2.76, CI 1.30–5.89). In further logistic regression models, GAS value and preoperative hematocrit values were found to be independent predictors for mortality (*p* = 0.023 and *p* = 0.007, respectively).
Korhonen, 2004 [43]	GAS	835	Univariate: coronary artery disease (*p* = 0.005), preoperative shock (*p* < 0.001), age (*p* < 0.001), and the GAS (*p* < 0.001). Multivariate: Preoperative shock [odds ratio [OR] 2.13 (CI 1.45–3.11); *p* < 0.001] and the GAS [for an increase of ten units: OR 1.81 (CI 1.54–2.12); *p* < 0.001]. The best cut-off value of the GAS in predicting postoperative death was 84 [AUC 0.75 (9% CI 0.72–0.78), standard error 0.17; *p* < 0.001].
Hsiang, 2001 [46]	Vancouver score	134	Preop > 90%, the sensitivity, specificity, and positive and negative predictive values were 25%, 98%, 95% and 54%, respectively. Mortality risk > 80%, values were 37%, 94%, 87% and 57%, respectively. Immediate postoperative mortality risk ≥ 90%; the sensitivity, specificity, and positive and negative predictive values were 17%, 87%, 60% and 49%, respectively. Mortality risk ≥80%; these values were 22%, 84%, 60% and 50%, respectively.
Tambyraja, 2004 [48]	ERAS vs. GAS, Hardman index, POSSUM and V-POSSUM	111	The GAS, Hardman Index and the ERAS were statistically related to mortality. However, the analysis via ROC curve revealed the ERAS to have an AUC of 0.72 (CI, 0.61–0.83). The V-POSSUM and POSSUM models had an ROC value of 0.70 (CI 0.59–0.82). The Hardman Index and GAS had an ROC value of 0.69 (CI 0.57–0.80) and 0.64 (CI 0.52–0.76), respectively.
Neilson, 2017 [51]	RrAAAS vs. GAS and ERAS	2704	Neither GAS nor ERAS provides a direct prediction of mortality; observed mortality in the VQI minus VSGNE cohort tended to be somewhat lower than predictions of the original RrAAAS. A recalibrated equation predicting the percent mortality was as follows: Mortality (%) = 16 + 12 × (age > 76) + 8 × (creatinine > 1.5) + 20 × (systolic blood pressure < 70).
Von Meijenfeldt, 2017 [52]	DAS	737	Age, lowest in-hospital systolic blood pressure, cardiopulmonary resuscitation, and hemoglobin level. ≥80% = 83%
Jàcome, 2021 [54]	Weingarten vs. GAS and Vancouver score	120	The three scores demonstrated some predictive value concerning mortality, although Glasgow Aneurysm Score demonstrated the highest area under the ROC curve (0.74) and the best discriminatory capacity for cut-off points with higher specificity. Neither of the scores demonstrated clinically useful predictive value.
Hemingway, 2018 [57]	HSR	118	Spreoperative risk score and subsequent 30-day mortality for all patients combined (*p* < 0.0001), for OAR patients alone (*p* = 0.0003) and for EVAR patients alone (*p* < 0.0001).

RR, risk ratio; CI, 95% confidence interval; OAR, open aortic repair.

## 4. Discussion

The use of predictive risk scores can be advantageous especially in emergent cases. They may guide and allow surgeons to tailor their approach and response strategies to the patients in a standardized way, as well as inform the relatives of the patients using objective data on what the predicted outcome might be and whether the procedure should be performed. Furthermore, risk scores may be used by physicians in general to evaluate patients before transport, and it can be arranged that these scores reach surgeons in a different hospital. Different authors have tried to compare and validate the various risk scores that are available, but due to the nature of the score themselves, a direct comparison may not always be feasible. Indeed, an inherent major limitation to performing comparative analysis to discern which clinical decision aid is ‘superior’ or ‘optimal’ is related to the significant heterogeneity of the different variables used among the different scores.

Hansen et al. [58] performed an accuracy evaluation using 38 patients for the DAS, the HSR and the VSGNE, with an ROC value of 0.762, 0.792 and 0.860, respectively, for all patients. When evaluating 30-day mortality for patients undergoing ruptured endovascular aneurysm repair, the scores were 0.802, 0.893 and 0.927, respectively. The difference between scores was not statistically significant. All three risk scores were significantly associated with the mortality rate using logistic regression. The authors concluded that each risk score can accurately predict 30-day mortality using the independent dataset. The results suggest that the VSGNE score is the most accurate; however, differences in accuracy between each scoring system were not statistically significant. The HRS scoring system is based only on preoperative variables. Although the VSGNE score had the highest ROC value in this analysis, it is dependent on intraoperative variables.

Van Beek et al. [59] performed a retrospective study in 49 patients in ten hospitals. The authors concluded that the most accurate mortality prediction came from the updated GAS for rAAA patients, while still not being reliable in classifying the futility of intervention.

Ciaramalla et al. [60] analyzed 49 patients who underwent surgery, including 33 patients receiving EVAR and 16 patients receiving OAR. The in-hospital mortality was 37% (24% for EVAR vs. 63% for OAR). Plots of the HRS and VSGNE scores vs. in-hospital mortality rate produced linear relationships (R^2^ = 0.97 and R^2^ = 0.93, respectively), in which a higher score was associated with a greater likelihood of mortality. Through logistic regression analysis using HRS score components, creatinine greater than 2.0 mg/dL produced a significant association with in-hospital mortality (OR 12.3; CI 1.1–131.7). 

Similar analysis using VSGNE components produced a significant association between suprarenal aortic control and in-hospital mortality (OR 5.5; CI 1.2–25.5). The ROC values were 0.74 (CI 0.60–0.88), 0.73 (CI 0.58–0.87) and 0.67 (CI 0.51–0.83) for the HRS, VSGNE, and DAS, respectively. The authors concluded that the HRS, VSGNE and DAS scores performed similarly and adequately predicted in-hospital mortality after rAAA. The HRS score holds the added benefit of using preoperative variables, setting it apart as a valid prognostic indicator in the preoperative setting.

Vos et al. [61] analyzed 347 consecutive patients. The AUCs were developed for the DAS, GAS, ERAS, Vancouver score and Hardman Index. The ROC value was better for the Vancouver score (0.716; CI 0.647–0.786) than for the other scoring systems. The ROC values for the DAS (0.664; CI 0.592–0.736), HI (0.664; CI 0.592–0.736), ERAS (0.621; CI 0.543–0.700) and GAS (0.591; CI 0.517–0.665) were slightly smaller, although only the difference between the Vancouver score and GAS was statistically significant. The calibration showed a good fit for all models. The authors concluded that the performance of the tested models for the prediction of mortality in rAAA patients was comparable, with only a statistically significant difference between the Vancouver score and the GAS in favor of the former.

More recently, Troisi and coworkers also analyzed and identified several intraoperative and pre-operative factors associated with in-hospital mortality [62] and long-term survival [63] (in those alive after 90 days from the index operation) for patients undergoing repair of an rAAA. Although they did not develop a specific risk prediction model, they were able to discriminate important variables (such as salvaging at least one hypogastric artery) that were likely missed by most registries before. Furthermore, they were also able to show that AAA-related death in the long run did not seem to be affected by the type of repair (open vs. endovascular), a finding that is similar to other recent studies [64]. Nonetheless, the higher peri-operative survival that is associated with EVAR over OAR [65] still mandates an endovascular-first approach in most rAAA patients [66], a trend that is confirmed by the recent literature [67].

To date, no vascular society has fully endorsed any of the risk scores available. This may be caused by the need for a risk score to be both fast and reliable in order to be clinically effective when dealing with emergent cases. Furthermore, a risk score cannot require specific training to be used or intra/postoperative data to evaluate the mortality risk of individual patients. If it must be used as a tool to decide which patients presents the higher possibility of survival or even if a repair may be futile, there can be no room for error. Unfortunately, the few comparisons between scores analyzed in the present scoping review do not give unequivocal results as to which score may be better suited for which situation. Further research needs to be made in order to truly understand which risk factors are to be taken into account and why. The best way to do this would be to create a worldwide multicenter registry to expand as much as possible the pool of patients.

Lastly, as technology is evolving, so should the vascular community and all stakeholders involved in the pathways of care for vascular patients. The use of machine learning and artificial intelligence may help to further improve decision making and risk stratification, as demonstrated in other surgical specialties [68,69]. The use of the ANN in vascular surgery to aid clinical practice [55] represents a pivotal step towards the implementation of such new technology, despite the need for external validation. The use of artificial intelligence and machine learning has already been endorsed in other surgical fields including emergency general surgery, and its implementation is also expected to rise in the vascular realm [70].

### Study Limitations

Findings from the present analysis should be interpreted within the context of some intrinsic limitations. A scoping review is an exploratory but systematic literature search that aims to address a broad topic and discover gaps in the evidence while providing a narrative review without any meta-analysis. As highlighted in the present analysis, the question(s) being addressed is usually broader and more complex and heterogeneous than that in a systematic review. However, specific gaps and unanswered questions in current literature can be identified by a scoping review and be addressed with new original research. Given the up-to-date methodological guidance and reporting criteria used for the present review, it could help direct future research on the mortality risk scores available in both major elective vascular surgery and emergent treatment of rAAA. At the same time, it can present the need for a more homogeneous reporting of data to make the cross-comparison of series and pooling of results feasible going forward. There may be an element of selection bias in the identification of articles for inclusion given the scoping nature of this review. Furthermore, pertinent articles may not have been found using the reported literature queries. The present work includes studies with heterogenous designs and methods, as well as a variety of definitions used for the postoperative outcomes assessed (e.g., 30-day vs. 90-day mortality). Also, center volume and physicians’ experience play a pivotal role in determining the outcomes of major vascular surgery and even more so for rAAA [71,72,73,74]; nonetheless, most risk scores fail to incorporate these elements in their prediction models, and how these observations would be translated from high-volume to low-volume institutions remains a matter to be investigated. Of the utmost importance is ascertaining how these mortality scores may be linked with possible intraoperative adverse events, a question that remains to be elucidated. In this regard, an ongoing global initiative (i.e., the ICARUS Global Surgical Collaboration) is currently underway to improve the definition and reporting of these adverse events, and therefore, it may possibly impact the diagnostic performance of the scores, which will need further external validation [75,76,77]. Lastly, we deliberated and subsequently elected to focus our paper on both “major vascular surgery” and “rAAA” (as the latter still represents a cumbersome clinical scenario for physicians and patients/families alike), although model validation may be inherently weaker for the latter owing to smaller population samples available for analyses. However, despite all this, the present scoping review represents a comprehensive assessment of a clinically relevant and complex subject, possibly guiding the direction of future research on this theme to improve the outcomes of major vascular surgery and for rAAA patients and to possibly understand when operations may be futile.

## 5. Conclusions

The present scoping review was able to detail an overview of the most commonly reported risk scores for predicting mortality in major vascular surgery patients and rAAA. Unfortunately, the published literature is characterized by significant heterogeneity of results among the various validation studies comparing the different risk scores. To date, no specific risk score has been endorsed by any vascular surgery society. As far as rAAA is concerned, an almost perfect prediction is needed to withhold intervention, and no existing scoring system is capable of achieving this requirement, although risk-scores can still be used to inform patients and caregivers regarding anticipated outcomes and to set expectations of care. Future developments in artificial intelligence may further assist and refine surgical decision making when attempting to predict post-operative adverse events more precisely.

## Figures and Tables

**Figure 1 jcm-12-05505-f001:**
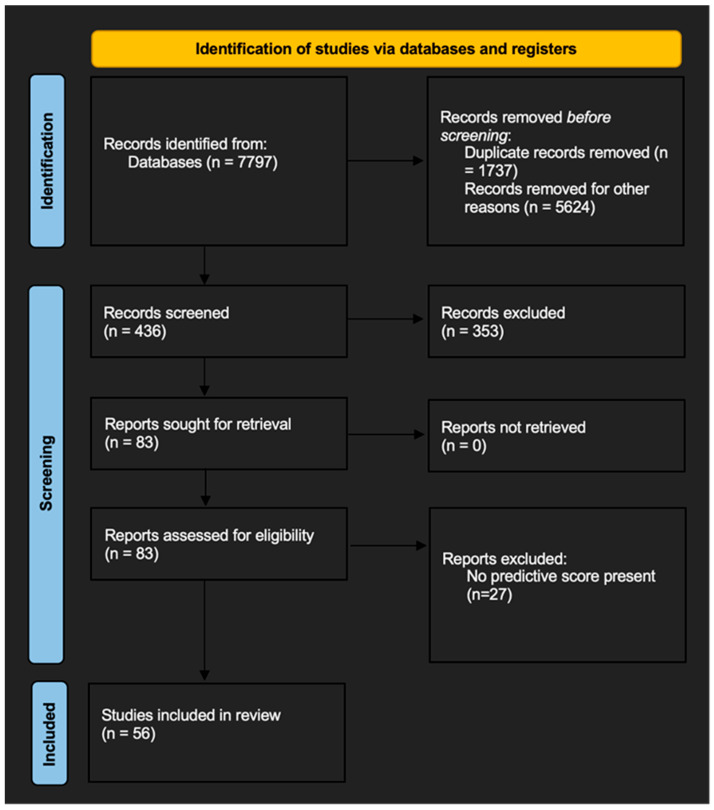
PRISMA flowchart of the literature selection.

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
