# Peer review of "Risk Prediction Models for Peri-Operative Mortality in Patients Undergoing Major Vascular Surgery with Particular Focus on Ruptured Abdominal Aortic Aneurysms: A Scoping Review"

_jcm, 2023, doi:10.3390/jcm12175505_

Round 1

Reviewer 1 Report

This PRISMA Scoping Review by Grandi et al is summarizing risk scores available in the literature for the prediction of peri-operative mortality during rAAA repair. The authors state that is difficult to make a comparison of these scores, due to the heterogeneity of variables used to build these scores. The use of artificial intelligence is needed to provide the best prediction for adverse events during rAAA repair. 

It is a relevant systematic scoping review since it attempts to compare different risk scoring systems used in rAAA with the ultimate goal to build a unique scoring system for peri-operative adverse events. 

Did the authors perform a search on Prospero to identify similar studies already available?

Their literature search leads to the identification of 56 studies to review. I could not find anywhere a clear list of these 56 studies. The number of studies that I counted in all the Tables is 34, not 56. Could you identify more precisely all the 56 studies and which score is determined in each study? 

Supplementary Table I is not clear. The page number is referred to which document? Are the numbers correct? 

In Figure 1, what is the meaning of the asterisks? 

Cited reference are relevant and most of them recent. The authors have included 10% of self-citated publications, which is acceptable.

the reading is fluid, although not perfect.

Author Response

Thank you for the comments. We hope that we were able to provide satisfactory answers to the reviewers' queries.

Reviewer 2 Report

The aim of the review was to compare the predictive accuracies of mortality risk scores in patients with rAAA. Moreover, the authors also collected general surgical risk scores. It is not clear what the latter had to do with the main topic of the review. I would recommend to focus on rAAA. As the authors mentioned, the existing evidence on these risk scores on rAAA remains spare, the same as for the most vascular surgical problems (excluding EVAR and carotid surgery). The methodology of the literature search cannot be evaluated because the supplementary table II which should contain the search strings, lacks in the present manuscript. The listing of the existing investigations revealed that not only the results but also the qualities were very heterogenous, whereby the case numbers reached from n=95 up to n>2000 (tables 3 and 4). How was the model cohort built, what was the method to get the risk score (probably multiple regression in most cases), how was the model evaluated by a test cohort? The statistical quality of the studies should be judged by the authors based on the fulltext articles. Furthermore, it is not clear how the predictive accuracy was quantified. The authors listed some values which were called "ROC", "ROC-value", "score" etc. As far as I know, the area under the curve (AUC) of the receiver operator characteristics (ROC) curve can be quantified as a measure of the predictive accuracy. Further parameters may be sensitivity, specificity, positive and negative predictive value.

The main problem of the manuscript seems to be the aim of the present literature review. What was the scientific purpose? We have a lot of risk scores which can, with more or less accuracies, predict that the more comorbidities and risk factors the rAAA patient brings along, the higher the risk of non-survival, an intuitive conclusion which is known to each surgeon from everyday's practice without any scores. It would have been rather more interesting what a surgeon can change in the management of rAAA in order to reduce lethality - EVAR instead of OS, preferably under local anesthesia, permissive hypotension, minimal interval from arrival till surgery etc. Regardings these aspects, the listed studies were not very informative. The authors suggested that these scores can also be used to exclude high risk patients from further treatment, and I recommend to do without such conclusions because of the ethical conflicts such verdicts may imply. Furthermore, endovascular methods are developing rapidly toward complicated anatomies and these scores may become outdated soon. There is also evidence that the success rate correlates with the caseload of the department or surgeon, and good expertise may be associated with much better results than those reflected in the scores.

The last but not least advice: the authors listed the PRISMA-checklist in supplementary table 1 but did not explain how they adhered (or not) to these standards. This should be part of a cautious, correct and balanced discussion.

Obviously, the authors come from different countries and institutions, an ideal precondition for using own data for evaluation or - even better - model building. If that had been already done before and cited here it should be clearly declared, otherwise, data can be collected and registered, and of course - the suggested multinational multicentre rAAA registry would be a very good idea and is worth being supported with data and knowledge.

I am not a native english speaker,  but I detected a lot of spelling mistakes and, therefore, recommend to either revise grammar and orthography thoroughly or send the manuscript to an editing service.

Author Response

(The authors gave the same response as above.)

Reviewer 3 Report

Dear Authors,

thank you for this work, which provides a comprehensive review of existing risk prediction models and their validation studies for patients undergoing major vascular surgery. The search for appropriate risk prediction models, especially to support decision making in life-threatening conditions such as ruptured AAA, is of continuing importance and interest. However, despite this interesting topic and the very detailed review, I would like to raise some points that need revision.

 My main concern is, that there should be a clear topic and direction within this work, even when it is considered a scoping review. Actually, it is not clear to me if all available scores for vascular surgery in general are in the focus, regardless of an evaluation for rAAA or if the authors aim for analyzing potential available scores regarding their ability in prediction outcomes especially for rAAA surgery. This is further complicated by the fact that, although the paper is intended to focus on rAAA according to its title, but not all of the discussed scores have been evaluated in (r)AAA populations. Therefore, I would recommend for rephrasing and focusing this paper strictly on scores with existing data derived exclusively from rAAA or at least AAA repair. Scores with a lack of specific rAAA data can be mentioned briefly for completeness but should be clearly separated and excluded from comparison regarding predictive value in rAAA repair surgery. Otherwise, the focus on rAAA should be abandoned and the scores should be compared regarding their predictive abilities within the entire field of (major) vascular surgery.

Some further/minor concerns are:

- validation for rAAA repair occurs often in smaller populations compared to score validation in major vasc. Surgery. This should be more emphasized when comparing the results.

- potential pitfalls in comparing scores with different endpoints (e.g. 30-day vs 90-day mortality) amongst validation studies should be considered and discussed accordingly

- Regarding the discussion, I agree with the authors, that predictive risks scores may be helpful for finding the adequate therapeutically approach in accordance with medical possibilities and the patient´s wish. But therefore we need highly validated scores which are exclusively based on preoperative variables. Referring to actual guidelines and there proposed endovascular first strategies, anatomy of the aneurysm, technical feasibility and availability of adequate grafts are the determining factors for choosing the initial therapy. Most of the here presented scores, with regard to the need of intraoperative/postoperative variables are only helpful in discussion and planning postoperative care. This lack for preoperative risk stratification should be emphasized and discussed more clearly.

-  LL72-74 sentence not clear…. Why is the requirement of data from supplemental testing crucial for the general ability of mortality prediction? I can follow the individual statements, but the context is not conclusive to me

- the script suffers from some typing and punctuation errors (esp. Page 12/13) and should be checked accordingly

- supplemental table 2 is missing

Author Response

(The authors gave the same response as above.)
